# Exploring the Interplay of Gut Microbiota and Systemic Inflammation in Pediatric Obstructive Sleep Apnea Syndrome and Its Impact on Blood Pressure Status: A Cross-Sectional Study

**DOI:** 10.3390/ijms252413344

**Published:** 2024-12-12

**Authors:** Chung-Guei Huang, Wan-Ni Lin, Li-Jen Hsin, Tuan-Jen Fang, Hsueh-Yu Li, Chin-Chia Lee, Li-Ang Lee

**Affiliations:** 1Department of Laboratory Medicine, Linkou Main Branch, Chang Gung Memorial Hospital, Taoyuan 33305, Taiwan; joyce@cgmh.org.tw; 2Research Center for Emerging Viral Infections, Department of Medical Biotechnology and Laboratory Science, Chang Gung University, Taoyuan 33302, Taiwan; 3Department of Otorhinolaryngology-Head and Neck Surgery, Linkou Main Branch, Chang Gung Memorial Hospital, Chang Gung University, Taoyuan 33305, Taiwan; wannilin@hotmail.com (W.-N.L.); lijen.hsin@gmail.com (L.-J.H.); fang3109@cgmh.org.tw (T.-J.F.); hyli38@cgmh.org.tw (H.-Y.L.); 4School of Medicine, College of Medicine, Chang Gung University, Taoyuan 33302, Taiwan; meiluosha@gmail.com; 5School of Medicine, College of Life Science and Medicine, National Tsing Hua University, Hsinchu 300044, Taiwan

**Keywords:** gut microbiota, systemic inflammation, blood pressure, obstructive sleep apnea syndrome, interleukin-17, tumor necrosis factor-α

## Abstract

Obstructive sleep apnea syndrome (OSAS) is prevalent among children and is associated with elevated blood pressure (BP), posing a risk for future hypertension and cardiovascular diseases. While the roles of gut microbiota and systemic inflammation in OSAS pathogenesis are recognized in adults and animal models, their impact on pediatric BP remains less understood. This cross-sectional study explored the relationships between polysomnographic parameters, gut microbiota, systemic inflammation, and BP in 60 children with OSAS. Significant associations between specific microbial profiles—including beta diversity and 31 marker microbes—and BP variations were observed. These microbial profiles correlated with significant alterations in systemic inflammation markers like interleukin-17 and tumor necrosis factor-α. Notably, the relative abundance of *Acinetobacter* was related to fluctuations in these inflammatory markers and BP levels. The research further highlighted the unique microbial and cytokine profiles exhibited by children with different BP levels, indicating a substantial role of gut microbiota and systemic inflammation in influencing pediatric cardiovascular health. The findings suggest integrating gut microbiota management into comprehensive cardiovascular risk strategies for children with OSAS. This initiative underscores the need for further investigations to decode the mechanisms behind these associations, which could lead to innovative treatments for pediatric OSAS.

## 1. Introduction

Pediatric obstructive sleep apnea syndrome (OSAS) is identified by frequent episodes of partial or complete upper airway obstruction during sleep [1], affecting approximately 1% to 5% of children aged 2 to 8 years [2]. This condition disrupts normal sleep patterns and leads to intermittent hypoxia, manifesting in symptoms such as habitual snoring, hyperactivity, and inattention [3]. Moreover, disrupted sleep can increase the presence of caries-causing bacteria and lower salivary pH and buffering capacity, potentially contributing to the development of dental caries in children [4]. Untreated, OSAS may result in growth retardation, learning difficulties, and cardiovascular complications [1,2,3]. Diagnosis involves a detailed medical history, physical examination, overnight polysomnography, and portable respiratory polygraphy [1], with the apnea–hypopnea index (AHI)—the number of apneas and hypopneas per hour of sleep—serving as a key diagnostic metric. An AHI ranging from one to three events per hour is generally considered normal, but diagnostic criteria may vary based on age, existing comorbidities, and other sleep study findings [5,6]. Treatment options depend on the severity of the condition and include lifestyle changes, surgical interventions such as adenotonsillectomy, and the use of continuous positive airway pressure therapy [7]. Prompt and appropriate treatment is essential to alleviate the negative impacts of OSAS on affected children.

The 2017 pediatric hypertension guidelines have refined the thresholds for diagnosing elevated blood pressure (BP) and hypertension in children, facilitating better early intervention strategies [8]. For children aged 1 to less than 13 years, BP classifications are specified as follows: normotensive BP is defined as below the 90th percentile; pre-hypertensive BP ranges between the 90th and less than the 95th percentile, or between 120/80 mmHg and less than the 95th percentile (whichever is lower); hypertensive BP is at or above the 95th percentile, or at or above 130/80 mmHg (whichever is lower) [8]. Elevated AHI during rapid eye movement (REM) sleep has been linked to morning hypertensive BP levels in adults with OSAS [9]. Moderate-to-severe OSAS is associated with increased systolic BP [10,11] and is acknowledged as a risk factor for future hypertension and cardiovascular diseases [12]. Although BP reductions are noted post-OSAS treatment, the correlation between these reductions and changes in OSAS severity shows variability [13]. The enduring effects of OSAS on BP, especially how these effects differ across various developmental stages, warrant further investigation [14].

The gut microbiota, a complex ecosystem within the gastrointestinal tract, plays a crucial role in systemic health and impacts conditions like pediatric OSAS and hypertension [15,16]. Changes in gut microbiota such as variations in alpha diversity (which measures the variety of microbes within individual samples [17]) and beta diversity (which assesses differences in microbial communities between samples [18]) can disrupt BP regulation and contribute to the development of hypertension [15,19]. Dysbiosis, defined as microbial imbalance, is associated with heightened systemic inflammation [20], which could elevate BP and increase cardiovascular risks in children with OSAS [16,21,22]. Conversely, beneficial microbes can produce anti-inflammatory substances like butyrate, which may enhance vascular health and potentially lower BP [23,24].

Inflammatory markers including interleukin (IL)-1β, IL-6, IL-17, interferon-γ, and tumor necrosis factor α (TNF-α) are associated with increased BP [25,26,27], while IL-10 is known to mitigate hypertension by safeguarding renal and vascular functions [25,27]. In children with OSAS, the sympathovagal balance during N3 sleep can influence the relationship between body mass index (BMI) and the serum level of IL-1 receptor antagonist [28]. For adults with both OSAS and hypertension, elevated markers of systemic inflammation (such as C-reactive protein and erythrocyte sedimentation rate) and abnormal metabolic profiles (notably elevated levels of blood glucose, uric acid, and magnesium) contrast with those seen in individuals with OSAS alone [29]. This complex interplay between the gut microbiota, systemic inflammation, and hypertension underscores the potential benefits of using probiotic and dietary interventions to foster a healthy microbiome and mitigate hypertension risk.

Furthermore, 16S ribosomal RNA (rRNA) sequencing, a critical tool in microbiome research, allows for detailed analysis of microbial diversity and has been instrumental in linking gut microbiota to systemic health [30,31]. Most previously published studies on gut microbiota, systemic inflammation, and BP regulation applied animal models and conducted investigations on adult populations. However, their interplays in pediatric populations, particularly in children with OSAS, are under-researched.

This study hypothesized that fecal 16S rRNA sequencing and serum inflammation biomarkers correlated with BP status in children with OSAS. The objective of this study was to elucidate specific gut microbiota and serum inflammation biomarkers associated with elevated BP in children with OSAS, aiming to identify potential targets for therapeutic intervention.

## 2. Results

### 2.1. Participant Characteristics

The cross-sectional study initially assessed 76 children with OSAS for eligibility and 66 children were recruited for examination of gut microbiota species analysis and serum inflammation biomarker measurement (Figure 1). After excluding six children lacking BP measurements, the final analysis comprised 60 participants: 13 girls (22%) and 47 boys (78%). These children had a median age of 7 years, with an age range between 6 and 10 years (interquartile range, IQR). The median body weight was 25.8 kg (IQR: 20.1–43.1 kg), median body height was 124 cm (IQR: 112–139 cm), median BMI was 17.4 kg/m^2^ (IQR: 15.2–23.0 kg/m^2^), and median AHI was 8.5 events per hour (IQR: 4.1–19.7 events per hour). The median systolic and diastolic BPs were 105 mmHg (IQR: 95–115 mmHg) and 65 mmHg (IQR: 59–71 mmHg), respectively.

### 2.2. Differences in Participant Characteristics and Polysomnography Parameters Across Three BP Groups

There were 38 (63%) children with normotensive BP, 12 (20%) with pre-hypertensive BP, and 10 (17%) with hypertensive BP (Table 1). The proportions of girls and boys were significantly different across three BP groups (*p* = 0.045). Except for systolic and diastolic BPs (both *p* < 0.001), no significant differences were observed across the three groups regarding the age, body weight, body height, BMI, AHI, apnea index (AI), arousal index (ArI), or mean and minimum peripheral oxygen saturation (SpO_2_).

### 2.3. Differences in Gut Microbiota Across Three BP Groups

The normotensive group identified 3636 OTUs, while the pre-hypertensive group had 2343 OTUs, and the hypertensive group had 1033 OTUs, with 609 OTUs shared among the groups (Figure 2a). The distributions of the top 20 genera were similar across the three groups, and the proportions of OTUs were comparable (*p* = 0.69) (Figure 2b).

Alpha diversity metrics, including the Chao1 and ACE indexes for community richness, along with the Shannon (Figure 3a) and Simpson indexes for community diversity, exhibited no significant differences across the groups, with all *p*-values exceeding 0.05. Additionally, Good’s coverage index indicated sufficient sequencing depth for accurate analysis, affirming the robustness of our data (Table 2).

In contrast, principal coordinates analysis revealed significant distinctions in gut microbiota composition between the normotensive and hypertensive groups, as evidenced by beta diversity assessments using the Bray–Curtis distance, which were statistically significant (*p* < 0.001) (Figure 3b,c).

LEfSe analysis, employing a linear discriminant analysis (LDA) score threshold of 2, identified 31 marker microbes associated with different BP levels, as demonstrated in the cladogram (Figure 4).

Notably, the pre-hypertensive group exhibited higher relative abundances of the genera *Roseburia*, *Lactobacillus*, *Erysipelotrichaceae UCG-003*, *Coprococcus 1*, and *Johnsonella*, as well as the species *Bacteroides coprocola DSM 17136*, *Streptococcus gallolyticus* subsp. *macedonicus*, and *Prevotella pallens*, compared to the other groups. In contrast, the hypertensive group showed increased relative abundances of the genera *Klebsiella*, *Gilliamella*, *Eubacterium ruminantium group*, *Snodgrassella*, *Aeromonas*, *Enterococcus*, *Paraprevotella*, *Bombella*, *Serratia*, and *Acinetobacter*, along with the species *Aeromonas veronii*, *Lactobacillus aviarius*, *Lactobacillus salivarius*, *Klebsiella variicola*, and *Serratia marcescens*, relative to the normotensive and pre-hypertensive groups.

### 2.4. Associations of Specific Gut Microbial Species with Participant Characteristics and Polysomnography Parameters

Significant inverse associations were observed between the relative abundance of *Ruminococcaceae UCG-003* and the AI (*r* = −0.40, *p* = 0.001), as well as between the relative abundance of *Bosea* and mean SpO_2_ (*r* = −0.38, *p* = 0.002) (Figure 5a). No significant associations were identified between the top 20 genera and the AHI, ArI, or minimum SpO_2_. Additionally, no significant associations were found between the top eight species and any polysomnographic parameters (Figure 5b).

### 2.5. Differences in Serum Inflammation Biomarkers Across Three BP Groups

Table 3 presents the distribution of hypertension-associated cytokine levels across the normotensive, pre-hypertensive, and hypertensive BP groups. There were significant variations in the levels of IL-17 and TNF-α among these groups. Notably, the pre-hypertensive BP group exhibited significantly lower IL-17 levels compared to the normotensive BP group, a finding that contrasts with previous studies [25,26,27], which generally report higher IL-17 levels associated with increased BP.

### 2.6. Associations Between Specific Gut Microbial Species and Serum Inflammation Biomarkers

Our analysis revealed several significant associations between the relative abundances of the top 20 genera and levels of hypertension-related serum inflammation biomarkers (Figure 5a). Specifically, *Acinetobacter* showed inverse relationships with IL-17 (*r* = −0.41, *p* = 0.001), TNF-α (*r* = −0.44, *p* = 0.0004), and interferon-γ (*r* = −0.40, *p* = 0.002); *Anaeroglobus* was inversely associated with IL-7 (*r* = −0.45, *p* = 0.0003), IL-17 (*r* = −0.35, *p* = 0.006), and TNF-α (*r* = −0.35, *p* = 0.006); *Bartonella* had inverse associations with IL-1β (*r* = −0.61, *p* < 0.0001), IL-6 (*r* = −0.35, *p* = 0.006), IL-17 (*r* = −0.55, *p* < 0.0001), interferon-γ (*r* = −0.45, *p* = 0.0004), and TNF-α (*r* = −0.54, *p* < 0.001); *GCA-900066225* was inversely related to IL-6 (*r* = −0.42, *p* = 0.0009), IL-17 (*r* = −0.41, *p* = 0.001), interferon-γ (*r* = −0.39, *p* = 0.0002), and TNF-α (*r* = −0.36, *p* = 0.004); and *Olsenella* showed inverse relationships with IL-1β (*r* = −0.45, *p* = 0.0003), IL-7 (*r* = −0.37, *p* = 0.001), IL-17 (*r* = −0.51, *p* < 0.0001), and TNF-α (*r* = −0.45, *p* = 0.0003).

Moreover, key findings from the top eight species indicated significant inverse correlations with cytokine levels (Figure 5b). Notably, *Apis mellifera* honey bee showed inverse relationships with IL-1β (*r* = −0.38, *p* = 0.003), IL-7 (*r* = −0.39, *p* = 0.002), IL-17 (*r* = −0.39, *p* = 0.002), and TNF-α (*r* = −0.35, *p* = 0.006); *Acinetobacter lwoffii* was inversely associated with IL-17 (*r* = −0.40, *p* = 0.002), interferon-γ (*r* = −0.40, *p* = 0.002), and TNF-α (*r* = −0.45, *p* = 0.0003); *bacterium NLAE-zl-c390* had an inverse association with IL-7 (*r* = −0.37, *p* = 0.003); *Bacteroides* sp. *SB5* was inversely related to TNF-α (*r* = −0.33, *p* = 0.0095); and *Lactobacillus mucosae* was inversely associated with IL-1β (*r* = −0.38, *p* = 0.003), IL-7 (*r* = −0.43, *p* = 0.0007), IL-17 (*r* = −0.49, *p* < 0.0001), and TNF-α (*r* = −0.41, *p* = 0.001).

## 3. Discussion

Our study underscores a significant correlation between pediatric OSAS and changes in gut microbiota, alongside systemic inflammation that varies with different BP levels. We identified distinctive microbial profiles, marked by beta diversity and 31 specific marker microbes, and two systemic inflammation biomarkers, characterized by IL-17 and TNF-α, which correlate with BP variations. Particularly, our analyses using LEfSe and Spearman correlations revealed notable associations with the genus *Acinetobacter*, related to both BP status and systemic inflammation. This finding, which both aligns with and challenges the existing literature, highlights the unique influence of *Acinetobacter*, warranting focused discussion within our results. Furthermore, other marker microbes did not show a direct relationship with hypertension-related inflammation biomarkers, suggesting their influence on BP might occur through different metabolic pathways or mechanisms. These results highlight profound disparities in microbial compositions and cytokine profiles among children across various BP categories, illustrating the intricate interactions between gut microbiota, systemic inflammation, and cardiovascular health in pediatric OSAS. The subsequent sections delve deeper into the potential mechanisms through which *Acinetobacter* and other microbes might influence BP regulation in this demographic.

### 3.1. Acinetobacter and Hypertension: The Role of IL-17 and TNF-α

*Acinetobacter*, a Gram-negative nonmotile organism comprising 84 child taxa [32], is commonly found in environmental settings like soil and water and is also present in the human gut microbiota [33]. This bacterium is known for its potential to promote systemic inflammation, a pivotal factor in the development of hypertension. Specifically, *Acinetobacter* can boost the production of endotoxins such as lipopolysaccharides (LPSs) [34], which are known to induce systemic inflammation and potentially impair vascular resistance and endothelial function [35]. Moreover, studies have linked an increased relative abundance of *Acinetobacter* in the peripheral blood microbiome with hypertension [36].

Contrary to typical expectations, our cohort data revealed that *Acinetobacter* presence correlated with decreased levels of inflammatory cytokines such as interferon-γ, IL-17, and TNF-α, with lower levels of IL-17 and TNF-α associated with higher BP (Figure 6). This observation is intriguing as it contrasts with prior studies [25,26,27], where typically, these cytokines are elevated in hypertensive conditions, suggesting a complex, context-dependent relationship between *Acinetobacter*, inflammation, and hypertension in pediatric OSAS.

During Acinetobacter baumannii infections, the bacterium is capable of triggering immune responses via dendritic cells, naive CD4^+^ T cells, and T helper 17 cells, thereby producing IL-17 and enhancing antimicrobial functions at gut barrier tissues [37]. This process is supported by mechanisms such as neutrophil recruitment and antimicrobial peptide production. Additionally, mast cells can initiate immune responses toward *Acinetobacter baumannii* by releasing TNF-α, activating effector neutrophils [38]. High levels of IL-17 and TNF-α might enhance barrier function and antimicrobial activity, potentially reducing *Acinetobacter* proliferation in relatively immunocompetent children with OSAS.

Our findings indicate that children with OSAS and normative BP exhibited higher serum levels of IL-17 and TNF-α compared to those with elevated BP. The roles of IL-17 and TNF-α in BP regulation involve various mechanisms, including endothelial function modulation, reactive oxygen species formation, vascular fibrosis, and renal sodium retention [39,40]. This underscores the necessity for further investigation into the complex interplay between gut microbiota, inflammatory cytokines, and BP regulation in pediatric OSAS, to better understand these dynamics and their implications for treatment and management strategies.

### 3.2. Gut Microbiota and Their Hypertension-Related Metabolites

Beyond the specific influences of *Acinetobacter*, broader gut microbiota relationships with hypertension are evident through various metabolites. Hypertension is associated with substantial alterations in gut microbiota composition and gut barrier function. This includes an increase in harmful bacteria and their detrimental metabolites such as hydrogen sulfide and lipopolysaccharides (LPSs), coupled with a reduction in beneficial bacteria that produce protective metabolites like short-chain fatty acids (SCFAs) [41]. Additionally, hypertension is associated with decreased intestinal tight junction proteins and increased intestinal permeability, which contribute to systemic inflammation and further complicate vascular health [42]. Individuals with hypertension typically show lower alpha diversity and reduced numbers of SCFA-producing microbiota, alongside a higher prevalence of Gram-negative bacteria that are primary producers of LPSs [43].

In our cohort of children with OSAS, those with elevated BP showed similar alpha diversity compared to their normotensive peers (Figure 3a, Table 2), yet they exhibited significantly lower beta diversity. This indicates that while the overall microbial richness may be similar, the evenness and distribution of microbial species differ significantly between hypertensive and normotensive children, suggesting that specific microbial profiles may be associated with hypertension in pediatric OSAS.

In children with OSAS, the higher relative abundances of five gut genera and three species were associated with pre-hypertensive BP status, whereas the higher relative abundances of ten gut genera and five species were related to hypertensive BP status (Figure 7).

Table 4 in our manuscript presents a detailed analysis of how specific gut microbial genera and species impact BP in children with OSAS. We categorize these microbiota into groups associated with pre-hypertensive and hypertensive BP states. For example, Gram-positive bacteria such as *Coprococcus 1*, *Enterococcus*, *Erysipelotrichaceae UCG-003*, *Eubacterium ruminantium group*, and *Roseburia* are linked with hypertensive conditions [44,45,46,47,48], likely through their roles in SCFA and butyrate production, lipid and immune–inflammatory modulation, and oxalate metabolism [49,50,51,52]. Conversely, *Johnsonella* and *Lactobacillus* are associated with BP reduction [53,54], potentially due to their involvement in dietary salt response, cholesterol metabolism, and the production of gut microbiota-derived metabolites such as trimethylamine-N-oxide and SCFAs [53,55].

Additionally, our findings underscore the significant influence of certain Gram-negative bacteria, such as *Klebsiella*, *Bacteroides coprocola DSM 17136*, *Acinetobacter*, *Paraprevotella*, *Serratia*, and *Bombella*, on hypertensive BP states in children with OSAS. These bacteria are known for their pro-inflammatory effects, which are mediated through various mechanisms including LPS pathways, SCFA production, stearoyl ethanolamide metabolism, responses to high salt intake, and oxidative stress [34,56,57,58,59,60,61]. The significant link of these Gram-negative bacteria with hypertensive BP [36,56,62,63,64] highlights their potential role in exacerbating hypertension. This detailed examination of gut microbiota interactions with BP regulation not only clarifies the complex dynamics within the microbiome of pediatric OSAS patients but also points to microbiota-targeted therapies as promising approaches for managing hypertension. The interplay between microbial-induced inflammation and hypertension suggests that modulating gut microbiota composition could become a strategic component in the comprehensive management of hypertension in pediatric OSAS, underscoring the need for further research to refine these therapeutic interventions.

Moreover, gut bacteria such as *Aeromonas*, *Aeromonas veronii*, *Bombella*, *Gilliamella*, *Klebsiella variicola*, *Lactobacillus aviarius*, *Lactobacillus salivarius*, *Prevotella pallens*, *Snodgrassella*, *Serratia marcescens*, and *Streptococcus gallolyticus* subsp. Macedonicus are involved in various biological processes including immune–inflammatory responses, lipid metabolism, LPS production, oxidative stress responses, and toxin production [55,60,61,65,66,67]. Despite their significant roles in these critical physiological and pathophysiological pathways, their direct relationships with hypertension have not yet been established in the scientific literature. Notably, species such as *Aeromonas*, *Aeromonas veronii*, and *Serratia marcescens* are known pathogens in humans that can lead to severe infections and shock [68,69,70], indicating their potential for significant systemic impact. This gap in understanding underscores the need for further research to elucidate the roles of these microorganisms in BP regulation, particularly in the context of their pathogenic capabilities and their interaction with host systems.

**Table 4 ijms-25-13344-t004:** Blood pressure (BP) regulation-associated gut microbial genera and species of children with obstructive sleep apnea syndrome.

Gut Microbiota	Classification	Possible Mechanism	Effect on BP in Human
Prehypertensive BP			
Genus			
*Coprococcus 1*	Gram-positive anaerobes	SCFA butyrate-related BP regulation [49]	Elevation in collegiate athletes with hypertension [44]
*Erysipelotrichaceae UCG-003*	Gram-positive anaerobes	Modulation of lipid metabolism and immune–inflammatory response [52]	Elevation in patients with hypertension [45]
*Roseburia*	Gram-positive anaerobes	SCFA butyrate-related BP regulation [50]	Elevation in patients with hypertension [46]
*Johnsonella*	Gram-negative anaerobes	Modulation of high salt diet response and immune–inflammatory response [53]	Reduction in humans with high salt intake induced high BP [53]
*Lactobacillus*	Gram-positive, aerotolerant anaerobes	Modulation of lipid cholesterol metabolism, immune–inflammatory response, oxidative stress response, GM-derived metabolites such as TMAO and SCFAs [55]	Reduction in patients with borderline hypertension [54]
Species			
*Bacteroides coprocola DSM 17136*	Gram-negative anaerobes	SCFA-related BP regulation [56]	Elevation in patients with hypertension [56]
*Streptococcus gallolyticus* subsp. *macedonicus*	Gram-positive facultative anaerobes	SCFA butyrate-related BP regulation [71]	Unknown
*Prevotella pallens*	Gram-negative anaerobes	Modulation of immune–inflammatory response [72]	Unknown
*Bacteroides coprocola DSM 17136*	Gram-negative anaerobes	SCFA-related BP regulation [56]	Elevation in patients with hypertension [56]
*Streptococcus gallolyticus* subsp. *macedonicus*	Gram-positive facultative anaerobes	SCFA butyrate-related BP regulation [71]	Unknown
Hypertensive BP			
Genus			
*Eubacterium ruminantium group*	Gram-positive anaerobes	SCFA butyrate-related BP regulation [51]	Elevation in patients with hypertensive disorder in pregnancy [47]
*Enterococcus*	Gram-positive facultative anaerobes	SCFA butyrate-related BP regulation, modulation of oxalate metabolism [51]	Elevation in patients with hypertension [48]
*Acinetobacter*	Gram-negative anaerobes	Modulation of immune–inflammatory response, GM-derived LPS [34]	Elevation in peripheral blood in patients with hypertension [36]
*Klebsiella*	Gram-negative facultative anaerobes	Modulation of immune–inflammatory response, GM-derived stearoyl ethanolamide [57]	Elevation in patients with hypertension [62]
*Paraprevotella*	Gram-negative anaerobes	Modulation of high salt diet response and immune–inflammatory response [58]	Elevation in patients with hypertension [63]
*Serratia*	Gram-negative facultative anaerobes	Modulation of immune–inflammatory response, GM-derived LPS and toxins [59]	Elevation in peripheral blood in patients with hypertension [64]
*Aeromonas*	Gram-negative facultative anaerobes	Modulation of immune–inflammatory response, GM-derived LPS and toxins [65]	Unknown; pathogens in human [68]
*Bombella*	Gram-negative aerobes	Modulation of immune–inflammatory response and oxidative stress response [60]	Unknown
*Gilliamella*	Gram-negative microaerophilic genus	Modulation of oxidative stress response [61]	Unknown
*Snodgrassella*	Gram-negative microaerophilic genus	Modulation of immune–inflammatory response and oxidative stress response [61]	Unknown
Species			
*Lactobacillus aviarius*	Gram-positive, aerotolerant anaerobes	Modulation of lipid cholesterol metabolism, immune–inflammatory response, oxidative stress response, GM-derived metabolites such as TMAO and SCFAs [55]	Unknown
*Lactobacillus salivarius*	Gram-positive, aerotolerant anaerobes	Modulation of lipid cholesterol metabolism, immune–inflammatory response, oxidative stress response, GM-derived metabolites such as TMAO and SCFAs [55]	Unknown
*Aeromonas veronii*	Gram-negative facultative anaerobes	Modulation of immune–inflammatory response, GM-derived LPS and toxins [65]	Unknown; pathogens in human [69]
*Klebsiella variicola*	Gram-negative facultative anaerobes	Modulation of immune–inflammatory response [66]	Unknown
*Serratia marcescens*	Gram-negative facultative anaerobes	Modulation of immune–inflammatory response, GM-derived LPS and toxins [67]	Unknown; pathogens in human [70]

Abbreviations: GM—gut microbiota; LPS—lipopolysaccharide; SCFA—short-chain fatty acid; TMAO—trimethylamine-N-oxide.

### 3.3. Study Limitations

While our study underscores the potential therapeutic impact of addressing interconnected factors such as gut microbiota and systemic inflammation in pediatric OSAS, several limitations merit consideration. Firstly, our relatively small sample size may restrict the generalizability of our findings, underscoring the need for a larger, more diverse cohort to validate these associations. Secondly, the observational nature of this study limits our ability to infer causality. Prior research indicated consistent alpha diversity across various OSAS severities but revealed significantly higher beta diversity in severe cases. Moreover, improvements in OSAS post-adenotonsillectomy were associated with changes in both alpha and beta diversities [73]. To conclusively determine the effects of microbiota-targeted interventions, such as dietary adjustments and the use of prebiotics or probiotics on BP, future research should implement randomized controlled trials. Thirdly, as BP measurements were conducted in a clinical setting, the possibility of white coat syndrome affecting our results cannot be dismissed. Incorporating 24 h ambulatory BP monitoring could provide a more accurate assessment of hypertension in subsequent studies. Lastly, while our research opens avenues for microbiota-targeted strategies as potential treatments or adjunct therapies for OSAS and elevated BP, further studies are essential to establish clear mechanistic links and determine how alterations in gut microbiota composition may directly influence BP regulation, potentially moderating pre-hypertension and hypertension in children with OSAS.

## 4. Materials and Methods

### 4.1. Study Design and Participants

This prospective cross-sectional study was conducted with approval from the Institutional Review Board of Chang Gung Medical Foundation, Taoyuan, Taiwan (Approval No.: 201507279A3). Written informed consent was obtained from both parents (full version) and participants aged 6 years or older (simplified version for children). The study conformed to the ethical standards of the Declaration of Helsinki and its amendments and adhered to the STROBE guidelines (version 4) [74,75].

Between March 2017 and January 2019, 76 patients diagnosed with OSAS were referred to the Department of Otolaryngology at Linkou Medical Center, Chang Gung Memorial Hospital, Taoyuan, Taiwan, for surgical evaluation [76]. The inclusion criteria targeted children aged 5–12 years with an AHI of ≥5.0 events/hour or an AHI of ≥2.0 events/hour accompanied by conditions such as elevated BP, daytime sleepiness, learning difficulties, growth failure, or enuresis [77]. Children with craniofacial abnormalities, neuromuscular disorders, or chronic inflammatory conditions, including asthma or autoimmune diseases, were excluded from the study [28,76,78].

Sixty-six participants were initially recruited after the inclusion and exclusion criteria were applied. Six were subsequently excluded due to the absence of available BP measurement data. The participant selection process is illustrated in Figure 1. Participants with acute inflammatory conditions or those undergoing antibiotic therapy were only eligible for blood and stool sample collections following a minimum two-week remission period [79].

### 4.2. BP Measurement and Categorization

Nocturnal BP was measured three times with a standard sphygmomanometer (Dinamap ProCare 100; GE Medical Systems Information Technologies, Inc., Milwaukee, WI, USA). Measurements were taken three times between 10:00 and 11:00 p.m., prior to the polysomnography exam. The children were seated quietly for 5 min beforehand, with their back supported, feet on the floor, and the right arm supported with the cubital fossa at heart level. The average of the three systolic and diastolic BP readings was calculated. If a reading indicated elevated BP (≥90th percentile) was observed, further measurements were taken using the auscultatory method. The BP measurement procedure was elaborated in previous publications [80,81]. For each child, age-, sex-, and height-adjusted percentiles for systolic and diastolic BP were determined [82]. Pediatric normotensive BP was defined as below the 90th percentile; pre-hypertensive BP as an average clinic systolic or diastolic BP between the 90th and less than the 95th percentile, or 120/80 mmHg to less than the 95th percentile (whichever is lower); and hypertensive BP as at or above the 95th percentile, or at or above 130/80 mmHg (whichever is lower), based on the child’s age, sex, and height [8,83].

### 4.3. Polysomnography

Full-night, in-lab polysomnography was performed using Nicolet Biomedical equipment (Nicolet Biomedical Inc., Madison, WI, USA) following the 2012 guidelines of the American Academy of Sleep Medicine [84]. The AHI was calculated by dividing the total number of apneas (defined as a ≥90% reduction in airflow lasting for ≥2 consecutive breaths) and hypopneas (defined as a ≥30% reduction in airflow associated with electroencephalographic arousal or a ≥3% reduction in SpO_2_ lasting for ≥2 consecutive breaths) by the total hours of sleep. This study evaluated several indicators of OSAS severity, including AHI, AI, ArI, mean SpO_2_, and minimum SpO_2_. Specific polysomnography procedures for this study are detailed in previous publications [76,79,85].

### 4.4. Fecal and Serum Sample Collection

Parents collected stool samples from participants in the morning and immediately snap-froze them before transporting them to the hospital [31]. Blood samples were collected at the hospital in the morning [79]. All samples were stored at −80 °C until analysis.

### 4.5. Fecal 16S rRNA Sequencing and Gut Microbiota Species Analysis

Genomic DNA was extracted from stool samples using a fecal DNA isolation kit (Mo Bio Laboratories, Carlsbad, CA, USA), with DNA quantity and quality assessed by a NanoPhotometer P360 (Implen GmbH, München, Germany). Amplification targeted the V3–V4 regions of the bacterial 16S rRNA gene using primers 341F and 806R, focusing on hypervariable regions to enhance resolution [86]. The PCR process followed Illumina’s 16S Metagenomic Sequencing Library Preparation protocol (Illumina, Inc., San Diego, CA, USA), using 12.5 ng of DNA, KAPA HiFi HotStart ReadyMix (Roche Diagnostics Corporation, Indianapolis, IN, USA), and a thermal cycling program: initial denaturation at 95 °C for 3 min; 25 cycles of 95 °C for 30 s, 55 °C for 30 s, and 72 °C for 30 s; with a final extension at 72 °C for 5 min.

Following PCR, product integrity was confirmed by agarose gel electrophoresis, with a target band of approximately 500 base pairs. Products were then purified using AMPure XP beads and subjected to a second PCR to incorporate Nextera XT Index Kit elements, including indices and adapters for Illumina sequencing. The quality of the indexed products was verified using a Qubit 4.0 Fluorometer (Thermo Fisher Scientific Inc., Waltham, MA, USA) and the Qsep100™ system before pooling for sequencing on the Illumina MiSeq platform, yielding paired 300-base pair reads.

Data processing included assembling paired-end reads using FLASH (version 1.2.11) [87], quality filtering with the QIIME pipeline (version 1.9.1) [88], and chimera checking with UCHIME (version 4.1) [89]. Operational taxonomic units (OTUs) were defined at 97% similarity using USEARCH (version 7.0.1090) [90]. Representative sequences were annotated with the Silva database (Release 132) [91] and aligned with PyNAST (version 1.2) against the Silva core set database [92].

### 4.6. Hypertension-Related Serum Inflammation Biomarker Measurement

Eight representative hypertension-related systemic inflammation biomarkers—IL-1β, IL-6, IL-7, IL-10, IL-15, IL-17, interferon-γ, and TNF-α [25,26,27]—were measured using a multiplex kit (Bio-Plex^®^ Pro Human Cytokine 27-plex panel, Bio-Rad Laboratories, Hercules, CA, USA). Whole blood was collected in the morning, and serum was separated from each sample. All procedures followed the manufacturer’s recommendations and have been previously reported [28,78,79]. Duplicate measurements for all samples were conducted using a Bio-Rad Bio-Plex Luminex 200 instrument and analyzed with Bio-Rad Bio-Plex Manager software (v6.0).

### 4.7. Outcome Variables

The primary outcome variable in this study was the composition of the gut microbiota. The secondary outcome variable was the measurement of hypertension-related serum inflammation biomarkers.

### 4.8. Statistical Analysis

Sample size determination (*n* = 58) was based on prior research [19], considering the Shannon index and ensuring adequate power for primary outcomes (effect size = 0.60; type I error = 0.05; power = 0.80).

Data analysis was performed using SPSS software version 29.0 (IBM Corp., Armonk, NY, USA) and R software (version 4.4.0) with the microeco package (v1.10.0) [93]. The Shapiro–Wilk test was used to assess normality, revealing that most continuous variables were not normally distributed. Descriptive statistics are presented as medians with IQRs for continuous variables and as counts with percentages for categorical variables. Group comparisons of continuous variables were conducted using independent-sample Kruskal–Wallis tests with Bonferroni correction applied for multiple comparisons. Spearman’s correlation test was used to assess correlations between study variables.

For microbial community analyses, the LEfSe method was used to visualize OTU intersections, while multivariate analysis of variance assessed microbial diversity. Significant taxa differences were identified with the LEfSe test, applying the Benjamani–Hochberg procedure to adjust for multiple comparisons [30]. Statistical significance was defined as a 2-tailed *p*-value of < 0.05, except for Spearman’s correlation test, where a stricter 2-tailed *p*-value of < 0.01 was used.

## 5. Conclusions

In this study, we established clear correlations between pediatric OSAS and significant changes in gut microbiota, directly linked with variations in systemic inflammation and different BP levels. Our findings identify distinct microbial profiles characterized by beta diversity and 31 marker microbes that correlate with specific BP levels. These correlations are particularly notable for their unique inverse relationships with inflammation markers like IL-17 and TNF-α, which differ from the typically observed patterns in hypertensive adults. This insight highlights a unique interplay between gut microbiota and cardiovascular health in pediatric OSAS, suggesting new potential therapeutic targets. We advocate further in-depth studies to fully understand these relationships and develop targeted treatments for this demographic, enhancing the management of cardiovascular risks in children with OSAS.

## Figures and Tables

**Figure 1 ijms-25-13344-f001:**
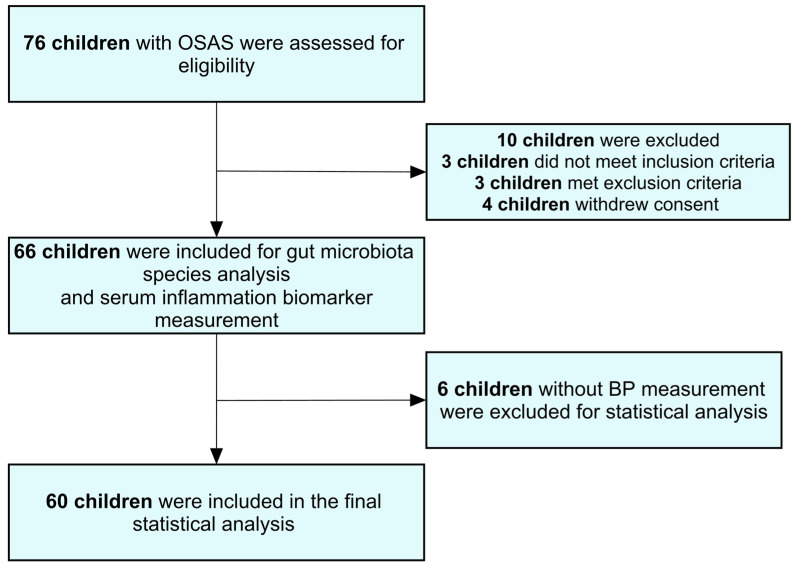
Flowchart of the present study. Abbreviations: BP—blood pressure; OSAS—obstructive sleep apnea syndrome.

**Figure 2 ijms-25-13344-f002:**
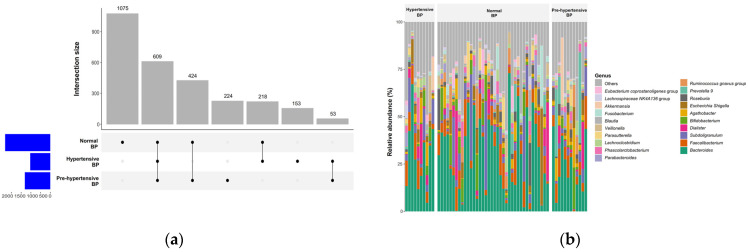
Comparison of normal blood pressure (BP), pre-hypertensive BP, and hypertensive BP groups. (**a**) The upset plot illustrates the intersections of operational taxonomic units (OTUs) among the three groups. Blue bars represent the number of unique OTUs in each group, while gray bars indicate shared OTUs across groups. (**b**) The bar plot displays the distribution of the top 20 genera among participants with different BP statuses.

**Figure 3 ijms-25-13344-f003:**
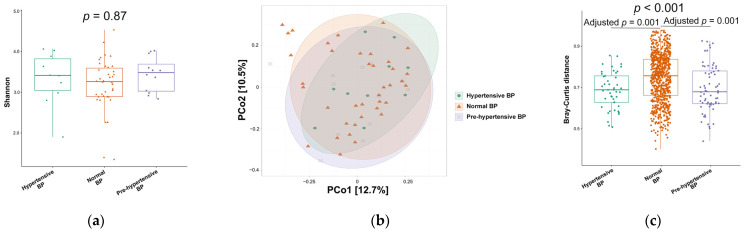
Differences in alpha and beta diversity across normal blood pressure (BP), pre-hypertensive BP, and hypertensive BP groups. (**a**) The box plot shows that Shannon index values, representing alpha diversity, were comparable across the three groups. (**b**) The principal coordinates (PCo) analysis plot illustrates the distribution of beta diversity among the three groups. (**c**) The box plot indicates significant differences in Bray–Curtis distances across the groups (*p* < 0.001), with the median Bray–Curtis distance in the normal BP group significantly higher than in the pre-hypertensive and hypertensive BP groups (both adjusted *p* = 0.001).

**Figure 4 ijms-25-13344-f004:**
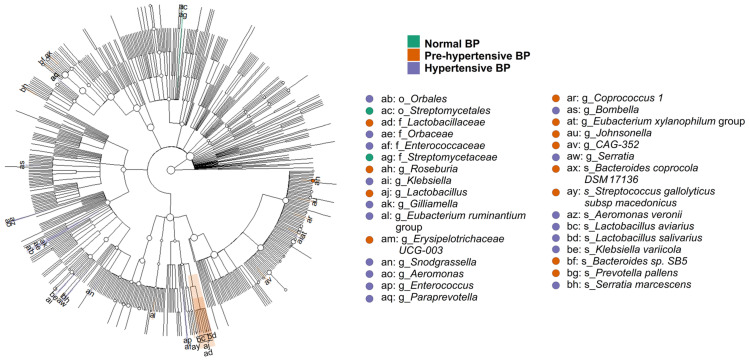
Cladogram of differential marker microbes across normal blood pressure (BP), pre-hypertensive BP, and hypertensive BP groups. Linear discriminant analysis of effect size with a score threshold of 2 identified 31 marker microbes associated with varying BP levels.

**Figure 5 ijms-25-13344-f005:**
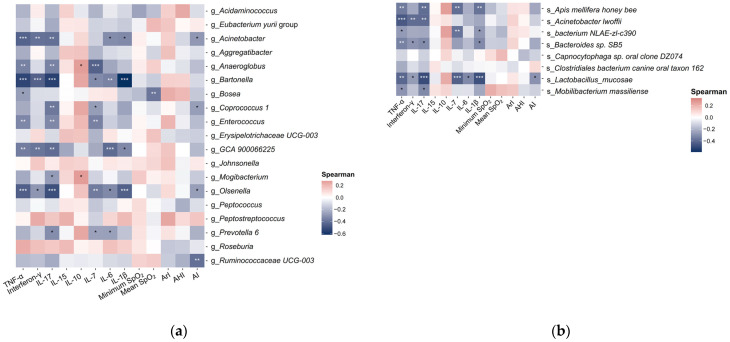
Associations between gut microbiota, polysomnographic parameters, and hypertension-related serum inflammation biomarkers. (**a**) Spearman’s correlations of the 20 representative genera with polysomnographic parameters and hypertension-related serum inflammation biomarkers. (**b**) Spearman’s correlations of the 8 representative species with polysomnographic parameters and hypertension-related serum inflammation biomarkers. Abbreviations: AHI—apnea–hypopnea index; AI—apnea index; ArI—arousal index; IL—interleukin; SpO_2_—peripheral oxygen saturation; TNF-α—tumor necrosis factor α. * *p* ≥ 0.01 and <0.05; ** *p* ≥ 0.001 and <0.01; *** *p* < 0.001.

**Figure 6 ijms-25-13344-f006:**
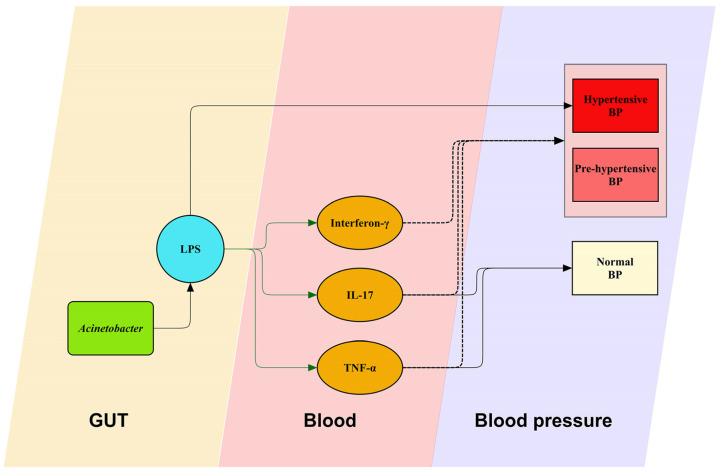
Mechanisms of blood pressure (BP) regulation by gut *Acinetobacter*. This figure illustrates the interactions between *Acinetobacter*, lipopolysaccharide (LPS), and inflammation biomarkers like interleukin-17 (IL-17), interferon-γ, and tumor necrosis factor α (TNF-α) in different BP groups. Solid black lines represent confirmed positive associations, green solid lines denote confirmed inverse associations, and black dashed lines highlight hypothetical positive associations that require further investigation.

**Figure 7 ijms-25-13344-f007:**
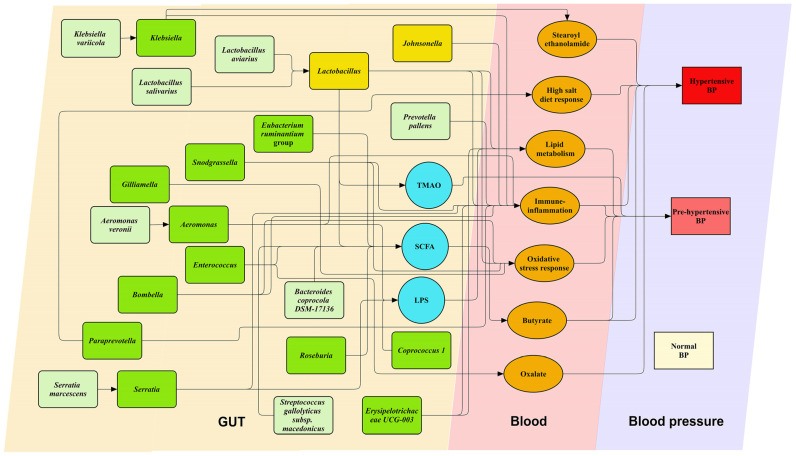
Gut microbiota and blood pressure (BP) regulation. This figure depicts the intricate relationships between various gut microbial genera and species and their association with BP categories. Green blocks represent microbial genera and species associated with elevated BP, while yellow blocks denote those linked to BP reduction. Black solid lines indicate significant associations between these microbial entities and BP changes. Abbreviations: LPS—lipopolysaccharide; SCFA—short-chain fatty acid; TMAO—trimethylamine-N-oxide.

**Table 1 ijms-25-13344-t001:** Clinical and polysomnographic data of children with obstructive sleep apnea syndrome across different blood pressure (BP) statuses.

Variables	Normotensive BP Group	Pre-Hypertensive BP Group	Hypertensive BP Group	*p*-Value ^a^
Clinical characteristics				
Subjects (girls/boys)	38 (12/26)	12 (1/11)	10 (0/10)	0.045
Age, years	6 (6–9)	8 (6–10)	8 (5–11)	0.42
Body weight, kg	24.4 (19.9–39.3)	28.5 (22.0–42.1)	32.5 (23.0–71.9)	0.30
Body height, cm	123 (110–131)	124 (113–140)	134 (113–155)	0.35
Body mass index, kg/m^2^	16.0 (15.2–21.9)	18.2 (16.2–22.9)	20.9 (15.0–30.2)	0.39
Systolic BP, mmHg	99 (89–106) ^b^	111 (105–115) ^b^	131 (117–138) ^b^	<0.001
Diastolic BP, mmHg	61 (58–65) ^b^	70 (64–75) ^b^	78 (74–88) ^b^	<0.001
Polysomnographic measures				
Apnea–hypopnea index, events/h	8.5 (4.0–17.7)	6.4 (4.1–16.9)	20.1 (4.4–26.1)	0.38
Apnea index, events/h	2.9 (1.3–5.3)	1.2 (0.5–2.9)	1.6 (0.9–2.3)	0.07
Arousal index, events/h	9.0 (7.1–16.6)	14.1 (7.6–17.3)	13.3 (9.4–23.6)	0.20
Mean SpO_2_, %	97 (97–98)	98 (97–98)	97 (96–98)	0.10
Minimum, SpO_2_, %	90 (86–92)	90 (83–92)	87 (83–91)	0.61

Continuous data are presented as medians (interquartile range); categorical data are expressed as numbers. ^a^ Comparisons between groups were performed using the independent-sample Kruskal–Wallis test with Bonferroni correction for continuous variables and Chi-square test for categorical variables. ^b^ The median difference between the normotensive and pre-hypertensive groups or the normotensive and hypertensive groups is significant (*p*-value < 0.05). Abbreviations: SpO_2_—peripheral oxygen saturation.

**Table 2 ijms-25-13344-t002:** Alpha diversity metrics of children with obstructive sleep apnea syndrome across different blood pressure (BP) statuses.

Variables	Normotensive BP Group	Pre-Hypertensive BP Group	Hypertensive BP Group	*p*-Value ^a^
Chao1 index, mean (median)	378 (286–680)	400 (301–443)	378 (238–461)	0.85
ACE index, mean (median)	366 (267–732)	404 (308–453)	372 (246–461)	0.75
Shannon index, mean (median)	3.56 (2.89–3.62)	3.48 (3.00–3.87)	3.42 (2.94–3.92)	0.43
Simpson index, mean (median)	0.91 (0.86–0.93)	0.91 (0.87–0.95)	0.92 (0.89–0.95)	0.55
Good’s coverage index, %, mean (median)	0.99 (0.98–0.99)	0.99 (0.99–0.99)	0.99 (0.99–0.99)	0.87

Continuous data are presented as medians (interquartile range). ^a^ Comparisons between groups were performed using the independent-sample Kruskal–Wallis test with Bonferroni correction for continuous variables.

**Table 3 ijms-25-13344-t003:** Hypertension-associated serum inflammation biomarkers of children with obstructive sleep apnea syndrome across different blood pressure (BP) statuses.

Variables	Normotensive BP Group	Pre-Hypertensive BP Group	Hypertensive BP Group	*p*-Value ^a^
IL-1β, pg/mL	0.72 (0.27–1.46)	0.22 (0.06–1.26)	0.29 (0.14–0.42)	0.09
IL-6, pg/mL	1.45 (0.68–2.36)	0.92 (0.17–1.58)	1.27 (0.87–4.42)	0.17
IL-7, pg/mL	7.55 (4.21–10.23)	5.56(1.72–9.64)	3.49 (1.72–9.64)	0.33
IL-10, pg/mL	0.39 (0.30–0.51)	0.51 (0.39–0.51)	0.39 (0.39–0.53)	0.72
IL-15, pg/mL	33.78 (7.79–66.01)	33.78 (15.97–33.78)	20.79 (7.79–40.48)	0.73
IL-17, pg/mL	18.40 (9.97–26.97) ^b^	5.20 (2.06–20.17) ^b^	11.88 (3.84–15.53)	0.01
Interferon-γ, pg/mL	3.45 (1.93–10.09)	2.46 (0.90–6.95)	2.32 (0.55–6.19)	0.24
TNF-α, pg/mL	41.65 (33.70–62.59)	30.68 (20.85–59.98)	34.50 (30.20–38.22)	0.04

Data are presented as medians (interquartile range). ^a^ Comparisons between groups were performed using the independent-sample Kruskal–Wallis test with Bonferroni correction. ^b^ The median difference between the normotensive and pre-hypertensive groups or the normotensive and hypertensive groups is significant (*p*-value < 0.05). Abbreviations: IL—interleukin; TNF-α—tumor necrosis factor α.

## Data Availability

The datasets generated and analyzed during the current study are available in the figshare repository, https://doi.org/10.6084/m9.figshare.22775159 [94].

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
