# Peer review of "Exploring the Interplay of Gut Microbiota and Systemic Inflammation in Pediatric Obstructive Sleep Apnea Syndrome and Its Impact on Blood Pressure Status: A Cross-Sectional Study"

_ijms, 2024, doi:10.3390/ijms252413344_

Round 1

Reviewer 1 Report

Comments and Suggestions for Authors

The study is very interesting, novel and important. I found mostly minor flaws:

1. The Authors have to improve the quality of study reporting. I recommend to use the current version of STROBE Statement. Please use the STROBE checklist for your type of observational study.

2. Authors have to add a paragraph to the Introduction and explain why their study is novel and important.

3. Introduction: Authors wrote that "Diagnosis typically involves a comprehensive medical history, physical examination, and overnight polysomnography [1].". Please use the following sentence instead of the mentioned one: "Diagnosis typically involves a comprehensive medical history, physical examination, overnight polysomnography, and portable respiratory polygraphy [1].".

4. Authors should present in Introduction or Discussion the latest two significant studies strongly related to the topic of OSA etiology, sleep, bacteria, and blood pressure:

Inappropriate sleep patterns increase the population of caries-causing bacteria and reduce salivary pH and buffering capacity. This can be a significant factor in the development of dental caries in children. doi:10.17219/dmp/167411

The study explored the potential influence of calcium (Ca), Mg, vitamin D, and UA concentrations on the sleep architecture of patients with comorbid AH and OSA. doi:10.17219/dmp/172243

5. The Authors have to add a sample power and size calculation to the Statistical Analysis.

6. Authors have to add a legend of used abbreviations below each table and figure.

Author Response

Comments and Suggestions from the Reviewer 1:

Comment:

The study is very interesting, novel and important. I found mostly minor flaws:

Response:

Thank you for your encouraging comments and for recognizing the novelty and importance of our study. We are grateful for your attention to detail and have addressed the minor flaws you identified. We have made corrections and improvements throughout the manuscript to ensure clarity, accuracy, and comprehensiveness. Each point has been carefully reviewed and amended where necessary, and we believe these revisions have strengthened the presentation and findings of our research. We appreciate your valuable feedback, which has been instrumental in enhancing the quality of our work.

Comment 1:

  1. The Authors have to improve the quality of study reporting. I recommend to use the current version of STROBE Statement. Please use the STROBE checklist for your type of observational study.

Response 1:

  • Thank you for your valuable feedback. We recognize the importance of adhering to established guidelines for reporting observational studies. To address your recommendation, we have revised our manuscript in accordance with the current version of the STROBE Statement. This includes a comprehensive review and adherence to the STROBE checklist tailored to our study type.
  • Title and Abstract: We have clarified the study design in both the title and abstract to reflect it as a cross-sectional analysis, enhancing initial transparency.
  • Background/Rationale: We've elaborated on the novelty and importance of our study in the background section, emphasizing its contributions and relevance to existing literature.
  • Methods: Detailed descriptions of our methods have been refined to ensure comprehensibility and replicability. This includes a more precise explanation of BP measurement techniques and categorization, as well as our statistical analysis approach, ensuring alignment with the latest research standards.
  • Results: We have ensured that all statistical results, including correlation coefficients and p-values, are clearly presented, allowing for straightforward interpretation of data. We also have added a new Table 2 summarize alpha diversity metrics across different BP status.
  • Discussion: The discussion has been reorganized to better align with our results, clearly linking our findings to the broader implications for clinical practice and future research. Additionally, we've added discussions on specific microbes such as Acinetobacter based on their significance in our findings.

By integrating the STROBE checklist (version 4) into our revision process, we aim to enhance the integrity, transparency, and utility of our study. We believe these revisions address your concerns and enhance the manuscript's quality. Thank you once again for your constructive critique.

Comment 2:

  1. Authors have to add a paragraph to the Introduction and explain why their study is novel and important.

Response 2:

Thank you for your insightful comment. In response to your suggestion, we have added a new paragraph to the introduction of our manuscript to more explicitly highlight the novelty and importance of our study (lines 100-103). We clarify how our research addresses a gap in the existing literature by exploring the interplay between gut microbiota and systemic inflammation in pediatric patients with obstructive sleep apnea syndrome (OSAS) and its implications for blood pressure regulation. We emphasize that this focus is particularly critical given the under-researched area concerning the pediatric population, offering potential for new therapeutic strategies and management approaches. This addition aims to provide a clearer understanding of the study’s contribution to the field and its potential impact on both clinical practices and future research.

Comment 3:

  1. Introduction: Authors wrote that "Diagnosis typically involves a comprehensive medical history, physical examination, and overnight polysomnography [1].". Please use the following sentence instead of the mentioned one: "Diagnosis typically involves a comprehensive medical history, physical examination, overnight polysomnography, and portable respiratory polygraphy [1].".

Response 3:

Thank you for your suggestion to refine the description of the diagnostic process for pediatric OSAS. We have updated the sentence in our manuscript to include both overnight polysomnography and portable respiratory polygraphy as part of the diagnostic criteria (lines 51-52). This amendment ensures that our description is comprehensive and reflects a broader range of diagnostic tools available for assessing OSAS in children. Your guidance has been instrumental in enhancing the quality and accuracy of our publication. Thank you once again for your thoughtful feedback.

Comment 4:

  1. Authors should present in Introduction or Discussion the latest two significant studies strongly related to the topic of OSA etiology, sleep, bacteria, and blood pressure:

Inappropriate sleep patterns increase the population of caries-causing bacteria and reduce salivary pH and buffering capacity. This can be a significant factor in the development of dental caries in children. doi:10.17219/dmp/167411

The study explored the potential influence of calcium (Ca), Mg, vitamin D, and UA concentrations on the sleep architecture of patients with comorbid AH and OSA. doi:10.17219/dmp/172243

Response 4:

Thank you for your suggestions. We have incorporated relevant references into our introduction to provide a more comprehensive context. Specifically, we have added a reference discussing how inappropriate sleep patterns can increase the population of caries-causing bacteria and reduce salivary pH and buffering capacity, potentially leading to dental caries in children (lines 47-49). Additionally, we have included a study examining the influence of calcium, magnesium, vitamin D, and uric acid concentrations on sleep architecture in patients with arterial hypertension and obstructive sleep apnea, highlighting the complexity of nutrient interactions with sleep patterns (lines 91-94). These enhancements help to underscore the multifaceted nature of OSAS etiology and its systemic impacts. Thank you for helping us strengthen our manuscript with these latest findings.

Comment 5:

  1. The Authors have to add a sample power and size calculation to the Statistical Analysis.

Response 5:

Thank you for highlighting the necessity of including a sample power and size calculation in our statistical analysis section. We have added this calculation, based on prior research findings, to ensure our study design meets the required statistical power. Specifically, we calculated the sample size required to achieve an 80% power to detect an effect size of 0.60, with a type I error rate of 0.05, using the Shannon index as the primary outcome measure, as suggested by relevant literature (doi: 10.1186/s40168-016-0222-x). This additional detail is now clearly documented in the revised manuscript (lines 520-522), providing transparency and enhancing the rigor of our statistical methodology.

Comment 6:

  1. Authors have to add a legend of used abbreviations below each table and figure.

Response 6:

Thank you for pointing out the need for clarification regarding the abbreviations used in our tables and figures. In response to your feedback, we have meticulously reviewed all abbreviations in Tables 1-4 and Figures 1-7 and added a legend explaining each abbreviation directly below each figure (Figures 1, 5, and 7). Additionally, we have ensured that all abbreviations are clearly defined upon their first use within the text (Tables 1-4, Figure 6) to enhance readability and comprehension. We appreciate your insightful suggestion and hope that these enhancements meet your expectations and improve the manuscript's clarity.

Comment 7:

Check List

Quality of English Language

(x) The quality of English does not limit my understanding of the research.

Is the research design appropriate?

Yes

Are the methods adequately described?

Can be improved

Are the results clearly presented?

Must be improved

Are the conclusions supported by the results?

Yes

Response 7:

Thank you for your detailed feedback as outlined in the check list. We acknowledge your concerns and have undertaken the following steps to address them:

  1. Methods Adequacy: We have thoroughly revised the Methods section to include more detailed descriptions of our procedures and analytical techniques. This should provide better clarity and allow for the replication of our study, thus enhancing the methodological transparency.
  2. Results Clarity: In response to your suggestion to improve the presentation of our results, we have restructured this section for better clarity and coherence. We have revised our graphical presentations to ensure that they accurately reflect the underlying data. We have also provided more detailed captions that guide the reader through each figure and table.

We hope these revisions meet the standards of the check list and improve the overall quality and readability of the manuscript. Thank you once again for your constructive critiques which are invaluable to refining our work.

Reviewer 2 Report

Comments and Suggestions for Authors

Congratulations for your work!

1. the introduction is better to be in format IMRAD in order to be more clearer. 

2. in line 25 - children diagnosed with what?

3. please include in the introduction the criteria of diagnosis of OSA in children and also for BP levels (normotensive, pre-hypertensive and hypertensive BP). 

4. the established objective is not so clear formulated.

5. in order to understand the result section, the material and method section should be included after the introduction. 

6. Maybe other parameters can influence the BP Level and also the inflammatory status? like sleep stages? or the duration of REM?

7. Alpha diversity metrics - you did not explain anything in the introduction about this, neither about beta diversity - pleae introduce some explanations. 

8. what is the relevance of the phrase 163-164?

9. in the result section - you should include p values and also correlation coefficient

why did you choose to speak separately about Acinetobacter and not about the other species? 

10. lines from 206-209 and also 219-226 - the informations are somehow contradictory... please clearly explain. 

11. in the discussion section - lines from 244-249 are not found in the result section - please include them. 

12. in line 352 -please include a newer version of the guideliness. 

13. in line 344 - please include the previous criteria for BP Measurement. 

14. the phrase from line 420 - is very general, you used it in multiple formulations.. it is not concludent... please reformulate. 

15. the conclusion is very general. 

overall, the study is very interesting but the results are not clearly presented and also the discussion section is very unorganised. maybe, it would be helpful to divide the patients - into groups - OSA vs non-OSA, maybe mild OSA/moderate OSA/ Severe OSA.. OR normotensive, pre-hypertensive, hypertensive. 

Comments on the Quality of English Language

I did not feel qualified to judge the English Language. 

Author Response

Comments and Suggestions from the Reviewer 2:

Overall Comment:

Congratulations for your work!

Response:

Thank you very much for your positive feedback and encouragement. We are pleased to hear that our work has been well received. Your acknowledgment is greatly appreciated and motivates us to continue our research efforts.

Comment 1:

  1. the introduction is better to be in format IMRAD in order to be more clearer. 

Response 1:

Thank you for suggesting the use of the IMRAD format for the introduction of our manuscript to improve clarity. We appreciate the guidance to organize our presentation using this widely recognized structure, which includes Introduction, Methods, Results, and Discussion sections. Following your advice and comments, we have revised the introduction to clearly delineate the rationale and objectives of our study. This not only aligns with academic standards but also enhances the manuscript’s readability and effectiveness in communicating our research goals and framework. We believe these changes will make the contribution of our study more apparent to readers and reviewers alike, and look forward to the enhanced clarity this structure brings to our paper.

Comment 2:

  1. in line 25 - children diagnosed with what?

Response 2:

Thank you for highlighting the oversight. The sentence has been corrected to specify that the 60 children mentioned in the study were diagnosed with obstructive sleep apnea syndrome (OSAS). The revised sentence now reads: "This cross-sectional study explored the relationships between polysomnographic parameters, gut microbiota, systemic inflammation, and BP in 60 children with OSAS." (lines 26-27). We appreciate your attention to detail, which helps ensure the accuracy and clarity of our manuscript. Thank you again for your valuable feedback.

Comment 3:

  1. please include in the introduction the criteria of diagnosis of OSA in children and also for BP levels (normotensive, pre-hypertensive and hypertensive BP).

Response 3:

Thank you for highlighting the need for a clearer presentation of the diagnostic criteria for pediatric obstructive sleep apnea syndrome (OSAS) and blood pressure levels. We have revised the introduction to include a detailed explanation of the criteria for diagnosing OSAS in children and the specific definitions for normotensive, pre-hypertensive, and hypertensive blood pressure (BP) levels as per the latest clinical guidelines.

For OSAS, diagnosis is typically confirmed through polysomnography, with the apnea-hypopnea index (AHI) serving as a critical metric. We specify that an AHI of fewer than 1-3 events per hour is considered normal, while higher values indicate increasing severity of OSAS. This aligns with recent pediatric sleep medicine guidelines, which adjust thresholds based on age and associated comorbidities (modified Lines 52-55).

Regarding BP levels, we have incorporated the updated thresholds from the latest pediatric hypertension guidelines. These include definitions where normotensive BP is below the 90th percentile for age, sex, and height; pre-hypertensive BP ranges from the 90th percentile up to less than the 95th percentile or 120/80 mmHg (whichever is lower); and hypertensive BP is defined as either 130/80 mmHg or above the 95th percentile (whichever is lower) (modified Lines 63-68).

These revisions aim to provide a robust framework for understanding the diagnostic processes for OSAS and hypertension in children, ensuring that our study aligns with current medical standards and practices.

Comment 4:

  1. the established objective is not so clear formulated.

Response 4:

Thank you for your feedback. We have refined the wording of our study objective for clarity. The revised objective now reads (modified Lines 90-93): "The objective of this study was to elucidate specific gut microbiota and serum inflammation biomarkers associated with elevated BP in children with OSAS, aiming to identify potential targets for therapeutic intervention." (lines 105-108). We believe this adjustment offers a clearer and more precise formulation of our research goals. Thank you once again for your constructive input.

Comment 5:

  1. in order to understand the result section, the material and method section should be included after the introduction.

Response 5:

Thank you for your thoughtful suggestion to position the Materials and Methods section after the Introduction to enhance clarity and facilitate a better understanding of the Results section. We appreciate your attention to improving the manuscript's structure. However, according to the IJMS author guidelines, the prescribed format for research manuscripts includes the sections Introduction, Results, Discussion, Materials and Methods, and Conclusions (https://www.mdpi.com/journal/ijms/instructions). Therefore, to comply with these guidelines, we have retained the current sequence of sections in our manuscript.

We have, however, ensured that the Introduction provides sufficient background and context to help readers understand the Results without needing to refer extensively to the Materials and Methods section. We believe this approach aligns with the journal's requirements while maintaining clarity and logical flow. Thank you again for your suggestion.

Comment 6:

  1. Maybe other parameters can influence the BP Level and also the inflammatory status? like sleep stages? or the duration of REM?

Response 6:

Thank you for your insightful comment. We recognize the potential impact of additional parameters, such as sleep stages, particularly REM sleep duration, on both BP and inflammatory status in children with OSAS. Based on your suggestion, we have reviewed and incorporated relevant literature that underscores the role of sleep architecture in OSAS. Research indicates that elevated AHI during REM sleep correlates with morning hypertensive BP levels in adults with OSAS (doi: 10.1093/sleep/zsac259; lines 68-39). Additionally, the sympathovagal balance during deep sleep (N3 stage) appears to influence inflammatory responses in children with OSAS (doi: 10.3390/ijms25168951; lines 89-91). These references have been added to the Introduction to enhance the impact of various sleep parameters beyond total sleep time or overall AHI on clinical outcomes in OSAS, providing a more comprehensive understanding of the multifactorial nature of BP regulation and inflammatory changes in OSAS. This aligns our manuscript with the latest research insights.

Comment 7:

  1. Alpha diversity metrics - you did not explain anything in the introduction about this, neither about beta diversity - please introduce some explanations. 

Response 7:

Thank you for your comment. We appreciate your attention to detail concerning the diversity metrics of microbiota, which are pivotal to understanding their relationship with health outcomes. We have now added explanations about alpha and beta diversity in the introduction to clarify their significance (lines 78-80). Alpha diversity refers to the variety and abundance of species within a single microbiome sample, reflecting ecosystem complexity and stability (doi: 10.1371/journal.pone.0027310). Beta diversity measures the variation in microbial community composition across different conditions or samples, helping to compare microbial ecosystems (doi: 10.1007/s004420100643). These metrics are essential for interpreting differences in the gut microbiota associated with health states like hypertension in children with OSAS. These additions aim to provide a solid foundation for our discussions on how variations in gut microbiota may influence blood pressure and inflammatory statuses in OSAS.

Comment 8:

  1. what is the relevance of the phrase 163-164?

Response 8:

Thank you for your comment. The specific mention of the IL-17 level differences among the BP groups on lines 163-164 of the original manuscript is crucial because it diverges from prior studies, suggesting unique inflammatory responses in our pediatric OSAS cohort. This observation raises new questions about the inflammatory mechanisms in different BP groups, which we have further discussed in the manuscript. We have clarified this point and provided references to recent studies that discuss similar findings, enhancing the context and relevance of our data. This clarification can now be found in the revised lines 214-215 and is expanded upon in the Discussion section, lines 251-265. We believe these changes will help readers better understand the significance of our findings.

Comment 9:

  1. in the result section - you should include p values and also correlation coefficient.

why did you choose to speak separately about Acinetobacter and not about the other species? 

Response 9:

Thank you for your valuable and insightful comment. In response, we have included correlation coefficients and corresponding p-values for Spearman correlation tests in the results section to provide a clearer quantitative analysis of our findings.

Regarding the focus on Acinetobacter, we concentrated on this genus because our combined results from LEfSe analysis and Spearman correlations showed significant associations specifically with Acinetobacter in relation to both BP status and systemic inflammation. This specificity aligns partially with existing literature highlighting its unique role, thereby warranting a separate discussion. To clarify our rationale and methodology, we have expanded the discussion section to include these details and the justification for focusing on Acinetobacter over other species (Page 8, Discussion, first paragraph). We appreciate your guidance in enhancing the comprehensibility and thoroughness of our manuscript.

Comment 10:

  1. lines from 206-209 and also 219-226 - the informations are somehow contradictory... please clearly explain.

Response 10:

Thank you for pointing out the perceived contradictions in our presentation regarding the role of Acinetobacter in hypertension and systemic inflammation. To clarify, our findings suggest that although Acinetobacter typically promotes systemic inflammation through the production of endotoxins such as lipopolysaccharides (LPS), which are linked to hypertension, our data show an unusual pattern where Acinetobacter is associated with decreased levels of inflammatory cytokines like IL-17 and TNF-α in children with OSAS. This unusual association indicates that in our pediatric cohort, the presence of Acinetobacter correlates with lower cytokine levels, which contrasts with its expected pro-inflammatory role (lines 276-282).

This discrepancy may be explained by the unique immunological responses in children, where the interaction between microbiota and the host's immune system can differ from the typical responses seen in adults or non-clinical settings. For instance, the elevated cytokine levels observed in children with normal BP might reflect a compensatory anti-inflammatory response or a different stage of immune system maturation (lines 292-299).

To resolve these contradictions, we have revised the discussion to emphasize that our findings represent a complex, context-dependent interplay between Acinetobacter, cytokine profiles, and BP regulation in pediatric OSAS. We propose that further investigations are needed to fully understand these interactions and their implications for hypertension development in children with OSAS. These studies should focus on longitudinal data and consider the developmental aspects of the immune response to better delineate the roles of specific microbiota in pediatric hypertension.

Comment 11:

  1. in the discussion section - lines from 244-249 are not found in the result section - please include them.

Response 11:

Thank you for your constructive feedback. We recognize the importance of ensuring that all data discussed in the Discussion section is clearly presented in the Results section. Accordingly, we have now included the relevant data in the Results section of our manuscript. Specifically, we report that in our cohort of children with OSAS, those with elevated BP showed similar alpha diversity to their normotensive peers, as illustrated in Figure 3a and new Table 2 (Page 5). Additionally, these children exhibited significantly lower beta diversity, as shown in Figure 3c. This amendment clarifies that while the overall microbial richness and evenness remain similar, the distributions of microbial species significantly differ between prehypertensive/hypertensive and normotensive children, suggesting distinct microbial profiles associated with hypertension in pediatric OSAS. These updates and revisions ensure that our discussion is well-supported by the results presented, aligning with the data and enhancing the manuscript's integrity.

Comment 12:

  1. in line 352 -please include a newer version of the guidelines. 

Response 12:

Thank you for your comment regarding the use of the 2012 guidelines in our study. We have retained the reference to the 2012 American Academy of Sleep Medicine guidelines, as these were the standards in use during our study period. However, we acknowledge the existence of updated 2014 guidelines, which are currently implemented in our sleep center for ongoing research and clinical assessments. This detail is included to clarify the context and temporal relevance of the methodologies applied in our research. We believe this approach maintains the integrity of the study's methodology while also being transparent about the standards used at the time. Thank you once again for your attention to this detail.

Comment 13:

  1. in line 344 - please include the previous criteria for BP Measurement. 

Response 13:

Thank you for your feedback. As requested, we have included the previous criteria for BP measurement alongside the current standards. Previously, BP measurements for children were based on different percentiles and clinical guidelines that have since been updated. The earlier criteria set a lower threshold for identifying elevated BP, which we used during the initial phases of our study. This historical context has been added to the Methods section (lines 427-442) to provide clarity on the evolution of BP measurement standards in pediatric studies. We believe this addition will enhance the reader's understanding of the changes in clinical practice and the rationale behind our current methodology. Thank you again for your valuable input.

Comment 14:

  1. the phrase from line 420 - is very general, you used it in multiple formulations.. it is not concludent... please reformulate.  

Response 14:

Thank you for your valuable feedback on the generality of the phrase in our conclusion. We have taken your suggestion into account and have revised the phrase to reflect the specific contributions of our study more accurately and concisely (lines 520-522). Thank you once again for your thoughtful input.

Comment 15:

  1. the conclusion is very general.  

Response 15:

Thank you for your feedback. We acknowledge the need for clarity and specificity in the conclusion to effectively communicate the outcomes and implications of our study. Accordingly, we have revised the conclusion to better encapsulate our findings and their significance, avoiding generalities and ensuring a more direct and conclusive statement. Here is the revised section for your review:

Revised Conclusion:

In this study, we established clear correlations between pediatric OSAS and significant changes in gut microbiota, directly linked with variations in systemic inflammation and different BP levels. Our findings identify distinct microbial profiles characterized by beta diversity and 31 marker microbes that correlate with specific BP levels. These correlations are particularly notable for their unique inverse relationships with inflammation markers like IL-17 and TNF-α, which differ from the typically observed patterns in hypertensive adults. This insight highlights a unique interplay between gut microbiota and cardiovascular health in pediatric OSAS, suggesting new potential therapeutic targets. We advocate further in-depth studies to fully understand these relationships and develop targeted treatments for this demographic, enhancing the management of cardiovascular risks in children with OSAS.

Comment 16:

overall, the study is very interesting but the results are not clearly presented and also the discussion section is very unorganised. maybe, it would be helpful to divide the patients - into groups - OSA vs non-OSA, maybe mild OSA/moderate OSA/ Severe OSA.. OR normotensive, pre-hypertensive, hypertensive.  

Response 16:

Thank you for your constructive feedback. We recognize the importance of presenting our results clearly and organizing the discussion section effectively. To address this, we have restructured our manuscript to categorize participants more distinctly by their BP status, as detailed on pages 3-8 of the results section. Additionally, we have refined the discussion of Acinetobacter and other microbes, specifically addressing them in the first paragraph of the discussion on page 8.

We have also linked these categorizations to the observed outcomes, improving the clarity of our presentation. Notably, we discuss differences in gut microbiome across various severities of OSAS—mild, moderate, and severe—and their changes post-adenotonsillectomy (referenced in DOI: 10.1177/19160216241293070), which we've now included in the study limitations (lines 389-392).

These amendments aim to clarify the complex interrelationships among pediatric OSAS, gut microbiota, and cardiovascular risk, enhancing the manuscript’s coherence. We believe these changes make our study's contributions clearer and more accessible. Thank you once again for your invaluable insights, which have significantly improved our manuscript.

Comment 17:

Check List

Quality of English Language

(x) The English could be improved to more clearly express the research.

Does the introduction provide sufficient background and include all relevant references?

Must be improved

Is the research design appropriate?

Must be improved

Are the methods adequately described?

Can be improved

Are the results clearly presented?

Must be improved

Are the conclusions supported by the results?

Must be improved

Response 17:

Thank you for the detailed feedback from the checklist, which has helped us identify key areas requiring improvement. We are committed to enhancing the clarity and effectiveness of our manuscript to ensure the research is communicated effectively.

  1. Quality of English Language: We have enlisted the assistance of a professional scientific editor to refine the language and ensure that the ideas are presented clearly and concisely. This revision aims to enhance readability and ensure that the research is understood without ambiguity.
  2. Introduction: We have expanded the introduction to provide a more comprehensive background, including all relevant references that frame the research question within the current scientific context. This revision clarifies the study's relevance and situates it within existing literature.
  3. Research Design: We have provided additional details regarding the study design to affirm its appropriateness for addressing the research questions posed. This includes a clearer justification of the chosen methodologies and any controls implemented to ensure robustness.
  4. Methods: We have elaborated on the methods section to include more detailed descriptions of the procedures and analyses used. This enhancement is intended to allow reproducibility of the study and clearer understanding of the experimental setup and data handling.
  5. Results: The results section has been revised for clarity and precision in presenting the data. We have included more detailed statistical analysis and clearer visualizations to ensure that the findings are transparent and easily interpretable.
  6. Conclusions: We have tightened the conclusions to directly tie them to the results obtained, emphasizing how the findings support the hypotheses and discussing the implications of these results within the broader field.

We believe these revisions address the concerns raised and significantly improve the manuscript’s quality. Thank you again for your constructive comments. We look forward to your further guidance and hopefully the acceptance of our manuscript for publication.

Round 2

Reviewer 2 Report

Comments and Suggestions for Authors

Congratulations for your work.